# Pandemic-Related Stress and Other Emotional Difficulties in a Sample of Men and Women Living in Romantic Relationships during the COVID-19 Pandemic

**DOI:** 10.3390/ijerph20042988

**Published:** 2023-02-08

**Authors:** Alicja Kozakiewicz, Zbigniew Izdebski, Maciej Białorudzki, Joanna Mazur

**Affiliations:** 1Department of Humanization of Health Care and Sexology, Collegium Medicum, University of Zielona Góra, 65-046 Zielona Gora, Poland; 2Department of Biomedical Aspects of Development and Sexology, Faculty of Education, Warsaw University, 00-561 Warsaw, Poland

**Keywords:** stress, relationships, love, COVID-19, women, men, gender inequalities, SEM models, indirect effects

## Abstract

This study examined the extent to which relationship quality affects variability in perceived stress and other emotional difficulties associated with the pandemic. The study was conducted 2–17 March 2022 using a self-administered online survey. The sample size consisted of 1405 individuals who were in a romantic relationship. The scales used in the study included the PSS-4, ECR-RS, SLS-12 and the standardized Pandemic-ED scale (RMSEA = 0.032). Increased stress levels (U = −5.741), pandemic-related emotional difficulties (U = −8.720), worse romantic relationship quality (U = −2.564) and more frequent anxiety-related attachment (U = −3.371) were characteristic of women. A hierarchical regression model for stress showed that age (b = −0.143), financial situation (b = 0.024), the ECR-RS scores (b = 0.219) and pandemic-related emotional difficulties (b = 0.358) proved to be statistically significant predictors of stress. The hierarchical regression model for pandemic-related emotional difficulties indicated five predictors: gender (b = 0.166), education (b = 0.071), financial situation (b = 0.203), scores on the ECR-RS scale (b = 0.048) and stress (b = 0.367). The SEM model used has satisfactory fit indices (RMSEA = 0.051), romantic relationship quality scores and attachment styles interact with the variability of perceived pandemic-related stress and burdens. The determined model offers conclusions relevant to clinicians working with individuals and couples during periods of intense stress.

## 1. Introduction

In response to the COVID-19 pandemic outbreak caused by the severe acute respiratory syndrome coronavirus 2 (SARS-CoV-2), the Polish government quickly introduced a lockdown, imposing restrictions and prohibitions in the interest of public health. People who at that time were not living together were strictly forbidden from seeing each other and physical contact became very dangerous. However, social distancing and mandatory home confinement were necessary to prevent infection. The global changes unleashed by the pandemic due to the spreading of the virus were inevitable, yet they had untold pressure on individuals and led to a common deterioration of public mental health [1,2,3].

Increased pandemic-related stress, life changes, social isolation and negative relationship quality were undoubtedly associated with worse mental well-being. Poorer social ties exacerbated the effects of the COVID-19 pandemic on people’s psychological well-being [4,5,6], triggering primary stressors, fear of infection and the related consequences on their own health and the health of their families, and secondary stressors, fear of social isolation, employment concerns and financial insecurity and the fear of a lack of resources [7]. Moreover, the impact of these stress factors on health and well-being was lower in individuals with high levels of self-control, self-esteem and/or social support [8,9]. Due to the burdens that the pandemic caused, positive coping strategies such as seeking social support, compassion, engaging in physical exercise, cognitive acceptance, avoidance of dreadful thoughts or positive thinking may have been helpful for many individuals [10,11,12,13]. One study showed that, during the COVID-19 pandemic, Polish students coped with stress mostly using the coping strategies of acceptance, planning and seeking emotional support [14]. Another study among Polish students also showed that the most commonly used coping strategies for stress during the second wave of the pandemic were acceptance, doing something else, active coping and physical activity [15].

Studies carried out before the pandemic showed that an excessive focus on social risks changes psychological processes that affect physiological functioning, impair sleep quality and increase morbidity and mortality [16,17,18]. Additionally, individuals with a lower quality relationship reported worse psychological well-being and a negative relationship quality is more strongly associated with poorer psychological well-being than a positive relationship quality [19,20].

Love is a universal emotion experienced by many people. Historical evidence suggests that people have experienced and have been fascinated by love across various historical eras and cultures worldwide. However, cultural conditions affect the way people feel, think and behave in a romantic relationship; so love is universal, but is also culturally specific [21]. In addition, the concept of love can mean different things in various types of relationships (e.g., friends, children and romantic relationships) and researchers have worked to create models that can distinguish between different experiences of love [22]. A romantic relationship is defined as a mutual, sustained and consensual interaction between two partners that is characterized by specific expressions of attachment and intimacy [23]. The need to belong and cultivate meaningful, positive interpersonal relationships is a basic human motivation and the satisfaction obtained from romantic relationships cannot be achieved through non-romantic relationships [24]. Moreover, how people interpret love plays an important role in predicting how satisfied they are in their relationship and, ultimately, whether it will survive [25].

During the COVID-19 pandemic, compared to single people, individuals living in a happy relationship reported better mental health than those with low relationship satisfaction and in a relationship with low commitment [26]. Here, it should be noted that romantic relationships and experiences are an important source of emotional bonding and contribute to the development of a positive self-concept and greater social integration [27,28]. Individuals in happy relationships showed greater subjective well-being than those in unhappy relationships, regardless of the relationship status [29]. Decades of research have demonstrated that people in romantic relationships are happier and enjoy better mental and physical health than unpartnered individuals [30,31,32]. Although the broad range of factors mentioned in the studies makes it difficult to establish what direct impact romantic relationships have on well-being, there is **a** general agreement in the literature that love is one of the aspects most strongly associated with personal happiness [33,34].

Moreover, the negative effects of COVID-19 compounded with the unintended consequences resulting from public health policies forced women to face problems ranging from COVID-19 infections and death, to long-term unemployment, to increased domestic violence not previously seen [35]. Undoubtedly, the COVID-19 pandemic is associated with significant health, racial and gender inequalities [36]. It is worth noting that front-line workers during the COVID-19 pandemic largely included women, who, more often than men, have caring responsibilities, which results from the horizontal and vertical segregation of the labor market and the fact that women tend to hold lower-paid jobs and are often associated with the social function of ‘care’ [37].

Although the severity and mortality associated with COVID-19 infection is twice as high in men than in women [38], women, compared to men, reported greater stress and anxiety at the beginning of the lockdown [39,40,41]. Household responsibilities increased for many individuals during the pandemic and gender inequalities were most vivid among those with children [42]. Women, particularly women in committed relationships with men, were expected to reduce their work time to take up care responsibilities, while, men were expected to do so in a lower extent, especially those living in committed relationships with women. It comes as no surprise then that women reported higher levels of stress and anxiety, as the pandemic increased both the burden of roles and expectations of women, and reduced, at the same time, external support [43].

Importantly, gender roles not only expect you to behave in a certain manner, but they are also predetermined, discouraging women from being cold and stoic and men from showing fear or anxiety. These gender roles became evident in the early stages of the pandemic, when women reported greater anxiety, stress and mental suffering, while men reported greater physical strength, peace of mind and determination [44].

### Research Question and Hypotheses

In this study the main research question was:

To what extent does the quality of the romantic relationship affect the variability of perceived stress and other emotional difficulties associated with the pandemic among men and women during the COVID-19 pandemic?

Based on the preceding literature review, we assumed the following:

**H1.** 
*Women experienced higher levels of stress and pandemic-related difficulties and rate their relationships more poorly than men.*


**H2.** 
*Negative relationship quality assessment and insecure attachment styles, relationship status, as well as socio-demographic variables affected the greater experience of stress and other pandemic-related difficulties.*


**H3.** 
*There was an indirect effect of a poor financial situation to increase stress for both men and women.*


**H4.** 
*Insecure attachment styles in relationships were associated with a higher level of stress directly and indirectly through the emotional response to the pandemic.*


In view of the knowledge available, this is the first such study in Poland that was conducted during the COVID-19 pandemic.

## 2. Materials and Methods

### 2.1. Participants and Procedure

The study was performed as part of a larger project on humanizing medicine and clinical communication from 2 to 17 March 2022. The sample size was N = 2050 and contained both adult patients and legal guardians of patients who received medical care between 2020 and 2022. Nevertheless, the sample accurately represents the population, as the use of health services over a 24-month period is highly likely. This survey used the technique of a self-administered online survey (CAWI) registered in a research panel provided by the Polish Research Collective Company. An individual link to an electronic survey form was sent with the survey invitation. Each respondent was double-checked for matching answers to questions about demographic information.

A survey instrument included questions about patients’ assessments of different dimensions of their relations with healthcare workers, as well as questions about certain areas of their personal lives and the influence of the COVID-19 pandemic on their evaluations. Participants provided answers to closed-ended questions, usually on nominal or ordinal scales. The questionnaire consisted of 296 variables. Only a small part was selected in this study. The average completion time was 28.8 min and the median value achieved was 23.9 min. Both values were calculated while taking into account only fully completed questionnaires.

Sample characteristics are shown in Table 1. To participate in the study, the respondents had to be in a formal or informal romantic relationship. A total of 1405 respondents met the inclusion criteria and were included in the sample for further investigation. The study sample is gender balanced, with 709 men and 696 women. The average age was 49.7 years (SD = 15.7). Most respondents had a university degree (39.6), followed by high school graduates (35.2%) and those with elementary and vocational education (25.2%). The sample is representative in terms of residence, with the greatest number of respondents residing in rural areas (38.2%), followed by those from small (31.2%) and big cities (30.6%). A vast majority of the respondents were in a formal relationship (marriage) (77.5%), while 22.5% were living in informal unions. The average length of the relationship was 21.9 years (SD = 15.9), and 83.6% of respondents had children. Concerning the financial situation, most respondents said their financial situation was the same (58.9%), 36.2% found it worse and 4.8% rated their situation as improved. The question the respondents were asked was "has your household financial situation deteriorated due to the COVID-19 pandemic?” and it referred to a change in the financial situation precisely during the pandemic.

### 2.2. Tools

The short four-item Perceived Stress Scale (PSS-4), a standardized measure also known as Cohen’s scale [45], was used as the first dependent variable. The scale is available in fourteen-, ten- and four-item versions. While the PSS-10 is highly recommended, some authors objected to the four-item version [46]. However, the latter works well in multithreaded questionnaires, as it allows for a reduction in the time needed to collect data or phone surveys. The PSS-4 comprises four questions asked from the perspective of the past month’s experience. An example of such questions is: in the past month, how often have you felt difficulties were piling up so high that you could not overcome them? In total, five categories of answers were provided, i.e., from never to very often. The answers were coded from 0 to 4 for negative statements and from 4 to 0 for positive statements. The summary index ranged from 0 to 16 points, where a high score meant a significant intensity of stress.

The second dependent variable used in the study was a scale that measured emotional difficulties experienced during the COVID-19 pandemic (the pandemic emotional difficulties scale). The respondents were asked to refer to the statement “because of the pandemic I feel…” with reference to four emotional states: frustration and/or uncertainty about the future, loneliness, anger and fear. The respondents used a five-point scale to rate their responses: never, rarely, sometimes, quite often and very often. Moreover, the respondents had the choice not to answer the questions. To test the psychometric properties, the analyses of the original pandemic emotional difficulties scale were carried out.

The Experiences in Close Relationships-Revised Scale (ECR-RS) was used as the independent variable. It is a nine-item tool for measuring attachment styles in different close relationships. The test–retest reliability of the individual scales was approximately 0.65 for the romantic relationship dimension and 0.80 for the parental dimension. Some of the ECR-RS items were: “It helps turn to people in times of need”; “I usually discuss my problems and concerns with this person”; “I talk things over with this person”; “I find it easy to depend on this person”; “I don’t feel comfortable opening up to this person”; “I prefer not to show this person how I feel deep down”; “I often worry that this person do not really care for me”; “I’m afraid that this person may abandon me”; “I worry that this person won’t care about me as much as I care about them” [47], s. 618. The permission of the author was received for the purposes of this study and the Polish translation by M. Marszal was used. The author of the tool provided the Polish translation at the request of the project team. The scale was a self-referential tool and included two dimensions, anxiety and avoidance. The same nine items were used to assess attachment styles in relation to four targets (i.e., mother, father, romantic partner and best friend). For the purposes of this survey, though, the assessment was measured only in relation to romantic partners.

Another independent variable used in the study was the SLS-12. Every item was presented on a five-point Likert scale, where 5 was “very much fits the description of my relationship” and 1 was “does not at all fit the description of my relationship”. The maximum score was 60 and the minimum score was 12. The total index assumed a range of 12–60. For both the total index and subscales, high scores suggest a stronger relationship. The SLS-12 scale includes 12 items and three subscales of two items, two items and eight items. Factor 1 related directly to sex life, factor 2 spoke of longing and affectionate gestures, while factor 3 focused on mutual respect, support, the capacity to resolve disagreements and a sense of security in a relationship. The overall Cronbach’s alpha coefficient for this scale was 0.959. Scores on the SLS-12 general scale ranging from 12 to 44 indicate a poor-quality relationship, scores ranging from 45 to 52 a moderately good relationship and scores of 53 to 60 a very good relationship [48]. Note that these two abovementioned scales were relevant for romantic relationships, yet the SLS-12 referred, to a greater extent, to a positive perception of a relationship and the ECR-RS identified attachment issues in a relationship.

The explanatory variables in the hierarchical regression and in the SEM model were the following: gender (0—men; 1—women); education (0—high school and lower; 1—university); place of residence (0—rural areas and small towns; 1—large and mid-sized cities); relationship status (0—informal; 1—formal); having children (0—no; 1—yes); and financial situation (0—other; 1—worse). Other variables were continuous or ordinal.

### 2.3. Statistical Analyses

As a first step, the distribution of total index values in the study sample of 1405 people was presented.

During the preliminary analysis phase, the psychometric properties of the pandemic difficulties scale were examined. The frequency distributions for the items and summary measures were described using mean and dispersion scores and the frequency of outliers. An item was considered to demonstrate a floor or ceiling effect if a large percentage of respondents were at the edges of the scale [49]. Up to 15% of effects were reported to be tolerable [50]. Skewness and kurtosis were both estimated to verify the normality of the data using item analysis and were checked with a multivariate normality test (n = 1405) using AMOS. Cronbach’s alpha coefficient was used to estimate the internal consistency of the data on the pandemic-related difficulties scale. Cronbach’s alpha values above 0.70 were generally expected to provide an indication of a reliable set of items. [51].

A maximum-likelihood estimation method was used to perform CFA along with 5000 bootstrapped samples due to violations of the assumption of multivariate normality. Bootstrapping is a resistant procedure for coping with non-normality in multivariate data [52,53,54]. The following model fit indices were shown as outcomes: CMIN/DF, comparative fit indices (CFI), goodness-of-fit index (GFI), adjusted goodness-of-fit index (AGFI) and root mean square error of approximation (RMSEA) and were used to evaluate the model’s fit. The values of the AGFI, the GFI, the CFI, the Tucker-Lewis index (TLI) and the normalized fit index (NFI) ≥0.90 indicated a good and adequate fit of the model to the data [55]. In confirmatory factor analysis, RMSEA values <0.08 were determined to be significant [56,57]. For CMIN/DF, a value lower than 5 indicated a reasonable fit, while a value of lower than 3 was recommended [58].

Next, Mann–Whitney U tests were used to calculate the group differences between men and women with regard to the scales under study. The non-parametric–parametric nature of the variables was tested using the Kolmogorov–Smirnov test procedure. The calculation of descriptive statistic parameters included means, standard deviations and percentages. The magnitude of differences between men and women was calculated by effect size analysis Glass’ delta. The rho Spearman correlation test was used to assess the correlation between individual scales. When comparing genders, we made comparisons between the scales and subscales (ECR-RS and SLS-12), whereas in correlation analysis we used general indices.

Hierarchical regression models were estimated, in which the dependent variables were stress and pandemic-related difficulties. Hierarchical regression analysis is a sequential investigation of the influence of multiple predictors, whereby the relative importance of a predictor is judged through incremental variance accounted by each predictor set [59]. The effect of the size of the associations is represented by the B coefficient.

Significant interactions between the analyzed factors were also sought using a general linear model (GLM). The results from this model are presented graphically as marginal means.

Furthermore, structural equation modeling (SEM) was used and a path model was estimated for the entire sample and for men and women. The maximum likelihood method and the following model goodness-of-fit statistics were used: TLI (Tucker-Lewis’s index), CFI (comparative fit index) and RMSEA (root mean square error of approximation) and SRMR (standardized root mean square residual).

To analyze the mediation effects between the said factors, the approach of Zhao et al. [60] was used, including the Monte Carlo method (bootstrap 5000 samples), to assess the standardized mediation effects with a 95% confidence interval [61].

In the SEM analysis, we included scales relevant to a romantic relationship, having children, the financial situation, education, age and pandemic-related emotional difficulties as factors that may affect the level of perceived stress during the COVID-19 pandemic and we developed a theoretical model used in further analyses (Figure 1).

## 3. Results

### 3.1. Pandemic-Related Emotional Difficulties Scale

As the pandemic ED scale is new, analyses directly related to the title of the paper were followed by an analysis of its structure and reliability. After analyzing the quality of fit of the model, the CMIN/DF value was showed to be 2.432, which indicated an acceptable result. The model’s absolute and incremental fit parameters demonstrated a good fit, as all values were >0.90 (CFI = 0.999, TLI = 0.996, NFI = 0.999, RFI = 0.994, incremental fit index (IFI) = 0.999, GFI = 0.999, AGFI = 0.991). The RMSEA and RMR values measured 0.032 (90%CI 0.001–0.007) and 0.008.

All the fit indices showed that the model fit was acceptable, as shown in Table 2. The Cronbach’s alpha for the entire tool was 0.844. The values of the standardized regression coefficients ranged from 0.65 to 0.87 (Figure A1). Table 3 presents descriptive statistics and floor and ceiling effects for individual items. Loneliness, anger and anxiety show right skewness with a maximum absolute value of 0.471. Frustration shows left skewness with a value of −0.056. The kurtosis coefficients ranged from −0.454 to −0.760. The analysis of the floor and ceiling effects shows that the proportion of extremely positive responses ranged from 5.4% to 12.7% and was the highest in frustration. The proportion of extremely negative responses ranged from 12.6% to 27.8% and was the lowest in frustration.

### 3.2. Correlational Analyses

Correlation analysis (Table 4) among the examined scales showed a moderate correlation between the stress scale and pandemic-related emotional difficulties (r = 0.444). Moreover, perceived stress correlated positively and moderately with the ECR-RS scale, while a poor and negative correlation was found between stress and the SLS-12 results (r = −0.248). A poor correlation was revealed between age and the level of perceived stress (r = −0.232), pandemic-related difficulties (r = −0.084) and attachment styles (−0.098). The correlation between relationship quality assessment and age appeared insignificant.

### 3.3. Differences between Women and Men in Stress and Pandemic-Related Difficulties

The obtained results, presented in Table 5, show there were no significant differences between women and men on the ECR-RS avoidance scores and SLS-12 sexual life and closeness scores. The ECR-RS scores were borderline significant. The mean score of perceived stress in women and men was 5.65 (SD = 3.17). Women achieved significantly higher scores (U = −5.741 *p* < 0.001), with a medium effect size (Glass’ Δ = 0.331). The mean score on the pandemic-related difficulties scale in both groups was 10.99 (3.83) and women achieved significantly higher scores (U = −8.720; *p* < 0.001); the effect size was large (Glass’ Δ = 0.474). The scores for anxiety were also statistically significant in women compared to men (U = −2.371; *p* = 0.018). Women, compared to men, were also characterized by a lower relationship quality (U = −2.564; *p* = 0.010)—effect size small (Glass’ Δ = 0.192) —and lower commitment (U = −3.386; *p* < 0.001)—effect size small (Glass’ Δ = 0.215). The results confirm that women and men differ in terms of experienced stress and emotional difficulties, as well as in relationship quality assessment and attachment pattern, with the score for the latter being borderline significant.

### 3.4. Determinants of Stress and Pandemic-Related Difficulties

We attempted to determine what factors affected the variability of the experienced stress and pandemic-related difficulties in patients receiving treatment during the COVID-19 pandemic. The results of the hierarchical linear regression models are shown in Table 6.

In model 3, pandemic-related emotional difficulties were found to be the most significant predictors of stress. The sociodemographic variables entered in step 1 and the love scale measurements accounted for 16% of the variance, while the ECR-RS measurement included in step 2 changed the accounted-for variance to 20%. The pandemic-related emotional difficulties variable, which was introduced in the final step, accounted for 30% of the total variance with respect to experienced stress. The last model consists of 11 predictors (factors), four of which, age, financial situation, scores on avoidance and anxiety in close relationships scales and pandemic-related difficulties, were found to be statistically significant.

Additionally, in the general linear model we looked for interactions among the analyzed factors. The assessment of attachment styles in close romantic relationships was transcoded at three levels (1: low anxiety and avoidance; 2: average anxiety and avoidance; 3: high anxiety and avoidance).

The interaction of those three factors was significant, at the level of *p* = 0.005. The analyses confirmed that the gender factor affects the perception of stress in interaction with other factors. Men reported lower stress levels, even considering the high scores on attachment styles. Both in men and women, the scores on perceived stress increased with the increase in scores on the ECR-RS, as shown in Figure 2.

Next, we considered a model with the independent variable of pandemic-related emotional difficulties, as shown in Table 7. In model 3, stress was the most significant predictor of pandemic-related difficulties. The sociodemographic variables and the love scale measurement entered in step 1 accounted for 17% of the variance, while the ECR-RS measurement included in step 2 changed the accounted-for variance to 28%. Perceived stress, the variable entered in the final step, accounted for 28% of the total variance with respect to pandemic-related difficulties.

The last model consisted of 11 predictors (factors), five of which—age, education, financial situation, scores on avoidance and anxiety in close relationships scales and stress—were statistically significant, while having children was borderline significant.

Additionally, in the general linear model we looked for interactions among the analyzed factors, as presented in Figure 3. The assessment of close romantic relationships was transcoded at three levels (1: low-quality relationship; 2: moderate-quality relationship; 3: high-quality relationship).

The interaction of those three factors was significant, at a level of *p* = 0.005. These analyses confirm that the gender factor affects the pandemic-related difficulties in interaction with other factors. Men reported fewer pandemic-related difficulties as their relationships improved, while in women this remained relatively stable, even with high scores on relationship quality. In men, romantic relationships proved to be a protective factor.

### 3.5. SEM Models

The study authors investigated the sources of stress variability during the COVID-19 pandemic and the SEM model was conducted to test whether the relationship quality and attachment pattern mediated the perceived stress.

The defined path model for the entire sample had satisfactory fit indices (RMSEA = 0.051). The TLI value was 0.949, CFI 0.969, while RMSEA was 0.051 and SRMR 0.041. Similarly, the defined path models for genders, men and women, also had satisfactory fit indices: for men TLS = 0.940, CFI = 0.964 and RMSEA = 0.055, SRMR = 0.042. and for women TLI = 0.947, CFI = 0.968, RMSEA 0.053, SRMR = 0.047.

Standardized paths from the SEM are presented in Table 8. As shown, the scores of the romantic relationship quality were significantly and negatively associated with the attachment pattern in a romantic relationship (b = −0.724, *p* < 0.01), whereas the attachment styles in a relationship were significantly and positively associated with perceived stress (b = 0.241, *p* < 0.001). Pandemic-related emotional difficulties were also significantly and positively (b = 0.378, *p* < 0.01) associated with stress. Having children also correlated significantly and positively with the perceived stress in the entire study sample (b = 0.094, *p* = 0.05) compared to the male sample, in which having children was not significantly associated with the perceived stress (b = 0.016 *p* = 0.650). Age was a significant predictor of the perceived stress in the entire study sample (b = −0.163, *p* < 0.001), as it was for both genders separately. The financial situation, as a mediating predictor, was significantly and positively associated with the presented attachment styles in the entire sample (b = 0.056, *p* = 0.02), whereas in women the financial situation was significant (b = 0.061, *p* = 0.015) and for men borderline significant (b = 0.051, *p* = 0.053). Note that education was not an important predictor in the entire study sample (b = 0.038, *p* = 0.123) or in men (b = 0.057; *p* = 0.103), whereas in women it was borderline significant (b = 0.067; *p* = 0.061).

The standardized indirect effects are shown in Table 9. The analysis of standardized indirect effects showed that finances were significantly and positively associated with pandemic-related emotional difficulties in the entire study sample (b = 0.013, *p* = 0.002) and women (b = 0.010, *p* = 0.010), while in men they were borderline significant (b = 0.014, *p* = 0.064). The analysis yielded similar results for the mediation effect of finances on stress, which was found to be significant and positive in the entire study sample (b = 0.018, *p* = 0.002) and in women (b = 0.020, *p* = 0.013), while in men the effect was borderline significant (b = 0.016, *p* = 0.068). The assessment of the relationship quality had a negative and significant effect on both pandemic-related emotional difficulties and perceived stress in all samples, at a level of *p* < 0.001, with the greatest indirect effect reported in men (b = −0.196) with respect to pandemic-related emotional difficulties and in women (b = −0.024) with respect to perceived stress levels. Education was not found to be significant in any study samples, whereas in women education was positively associated with perceived stress with borderline significance (b = 0.024, *p* = 0.055). Attachment styles in a romantic relationship were positive and significant across all study samples (*p* < 0.001), with the highest standardized indirect effect in men (b = 0.103).

The study results prove that the relationship quality assessment and attachment styles affected the variability of perceived stress and pandemic-related difficulties and that relationship attachment styles affected perceived stress levels directly and indirectly via an emotional reaction to the pandemic. Given hypothesis four, our study results confirm that financial factors have a strong indirect effect on stress through attachment styles in both study samples.

## 4. Discussion

In response to the outbreak of the COVID-19 pandemic, specific regulations and restrictions, as well as the associated uncertainties, resulted in increased emotional stress in the population [62]. The literature review clearly shows that individuals across the world reported worse mental well-being and greater depression and anxiety than before the pandemic [63,64]. In times of prolonged stress and challenges, interpersonal relationships become increasingly important [65] and in crisis situations, people tend to turn to and rely on their close ones and in adulthood, these are most often their romantic partners [66].

Research has shown that romantic relationships are associated in a particular way with subjective well-being. Marriage has been cited as one of the main sources of both support and stress. In both genders, support from the romantic partner and family were predictors of well-being, whereas partner strain was predictive of health problems [67]. Marriage is particularly associated with lower levels of psychological suffering and greater well-being in adulthood [68,69].

Regarding hypothesis one, this study showed statistically significant differences between men and women living in romantic relationships with respect to perceived stress, pandemic-related difficulties and relationship quality assessment and attachment styles (anxious and avoidant), with women reporting greater stress and emotional difficulties and less favorably assessing their relationships.

Women appeared particularly vulnerable to the negative effects of the COVID-19 pandemic and our study results confirm the study results obtained in Great Britain, where women reported more mental health problems than men at the time of the pandemic [70,71]. Moreover, one study showed that the decline in mental well-being among the UK respondents was twice as large for women as it was for men [72]. A study conducted in 26 countries showed that women experienced greater increased stress than men during quarantine caused by the COVID-19 pandemic [73]. A cross-sectional study with respondents from Germany and Austria, conducted during the COVID-19 pandemic, found that single, unmarried or unpartnered younger women experienced greater stress [74]. Furthermore, another study suggests that gender inequalities contributed to lower indices of sexual functioning and satisfaction and may have deepened the pleasure gap between men and women. Importantly, in our study, women also reported greater stress than men and a greater emotional burden in that respect [75].

In the study sample, women had higher scores on the anxiety attachment style and were less committed in their relationships. Despite being statistically insignificant, the mean scores of the sexual life and relationship closeness subscales were lower in women. Research shows that both men and women experienced more stress during the pandemic than before because they felt restricted in their relationships. However, women reported significantly more perceived relationship stress in lockdown than before the pandemic due to conflicts in their relationships [76].

Men experienced less stress—even considering high scores on attachment styles—with both men and women with insecure attachment styles experiencing more stress, which confirms that early nonadaptive attachment experiences provide inadequate and unstable stress regulation mechanisms, hindering thereby the development of the psychological resources necessary to cope with stressful events in life [77]. Differences between partners’ attachment styles may have aggravated difficulties in coping with and adapting to the COVID-19 crisis.

Moreover, a romantic relationship proved to be a protective factor for men; as their relationships improved, men reported fewer pandemic-related difficulties, which was not the case in women. It should be noted, however, that women, compared to men, paid more attention to the relationship status [78]. Also, women typically experienced more chronic and daily stress compared to men [79] and were less satisfied with their marriage [80]. Similar differences emerged in a study, which highlights that women are less satisfied with their marriage and life compared to men [81].

On the other hand, Randall et al. [82] found that in the Bangladeshi, Canadian, Chilean, Ghanese and Spanish subjects, positive dyadic coping did not mitigate the association between mental suffering after COVID-19 and relationship quality, which we also found was true for women in our study. They also highlighted that they did not find clear differences between those countries in terms of financial resources, governments’ responses, the scope of the pandemic or other cultural aspects that would account for why positive dyadic coping did not moderate the association between mental suffering and relationship quality [82]. It was also reported that COVID-19-related concerns significantly endangered mental well-being and unhappy partners were more at risk than happy partners, as they had fewer interpersonal resources and showed lower levels of individual well-being [83].

With regard to the second hypothesis, the hierarchical regression model for stress as an independent variable showed that age, financial status, the scores on the avoidance and anxiety scales in close relationships and pandemic-related emotional difficulties were statistically significant predictors of stress. While the hierarchical regression model for emotional difficulties as an independent variable revealed six predictors for difficulties—age, education, children (borderline significance), financial situation, the scores on avoidance and anxiety scales in close relationships and stress—that were statistically significant. Both final models did not confirm the importance of assessing the quality of a romantic relationship in terms of experiencing the aforementioned difficulties.

Considering age as a predictor for stress, numerous studies confirm that COVID-19-related mental stress was higher in younger adults [84,85]. While most studies focused on the general population, the COVID-19 pandemic produced new challenges and opportunities for families that may have impacted the mental distress of parents and consequently children’s well-being [86,87].

A growing number of studies into the impact of the COVID-19 pandemic on families stress the importance of considering individual risk as well as the interaction of parental (couple) and pedagogical (children’s well-being) factors [88,89]. Particularly, recent findings revealed that parents tended to be more stressed than nonparents during the COVID-19 pandemic [90,91]. Moreover, for mothers, personal mental issues and having younger children were found to be predictors for greater parental exhaustion [92], as confirmed by our findings—in the SEM model having children predicted stress only in women—and by studies, where mothers reported a poorer work balance compared to fathers and rated the balance between their work and family and doing things for others versus doing things for themselves less favorably [93].

The study also showed that finances matter a great deal for various aspects of couple and family relationships [94,95]. Finances are consistently identified as one of the major stressors for both individuals and relationships [96]. The COVID-19 pandemic together with the associated lockdown and restrictions created global economic challenges, resulting in increased financial stress among many individuals. A demand for families during the COVID-19 pandemic was financial stress (i.e., a growing number of financial stressors such as the loss of a source of income and housing insecurity because of the pandemic). Some respondents reported increased stress in their relationships and demands resulting from COVID-19-related financial stress [97].

This study shows that finances were a significant predictor of stress and other pandemic-related difficulties. In the standardized model, finances indirectly affected both stress and the pandemic-related emotional difficulties; however, these effects were statistically significant only for the general population and women. Therefore, the third hypothesis has been partially confirmed. One study [98] revealed that it may be assumed that women, due to significant work-related difficulties, worried more about finances than men. Yet another study [99] found that women were more susceptible to the devastating impact of the COVID-19 pandemic. Further, women worldwide were more vulnerable to stress, anxiety and depression and job loss during the pandemic [100]. And a study found that perceived stress, but not economic pressure or pandemic concerns, was associated with relationship instability [101]. Undoubtedly, the impact of finances—broadly understood as a socioeconomic situation—requires further investigation, whilst bearing in mind that, although the lifestyle of people with a more stable financial condition might have changed during the COVID-19 pandemic, the pandemic meant greater instability for individuals already affected by financial hardship, which in turn probably resulted in greater stress.

These study findings show the importance of romantic relationships in the lives of individuals, both in men and women, and the potential sources of risks and resilience that shaped the perception of stress during the COVID-19 pandemic. Regarding hypothesis four, the relationship quality and attachment styles in romantic relationships affected, directly and indirectly, stress levels in both men and women.

Importantly, the most powerful path of indirect effects was the relationship quality assessment as a predictor for lower sexual, romantic and individual functioning on both levels of the SEM. Increased stress related to COVID-19 during specific months predicted relevant declines in sexual functioning, which in turn predicted declines in individual well-being [102]. Another study supports our findings, showing that mental well-being was worse among individuals in relationships with low commitment and those living with a child, as compared to relationships with high commitment or without a child. Our findings showed that both having children and relational attachment styles were associated with perceived stress during the COVID-19 pandemic [26].

These findings are consistent with previous findings, suggesting that experiencing sudden and significant stress may negatively affect a relationship [103,104]. It is worth noting that, according to the systemic transactional model [105], romantic partners play an important role in mutual stress management when individual resources have been exhausted, as might have been the case during the COVID-19 pandemic. Moreover, it is considered that conflicts are destructive moderate associations between relationship quality and emotional health across all relationships—a romantic or family relationship or friendship. Tension in a relationship particularly impacts the well-being of individuals who believe that conflict is debilitating rather than productive [106].

For the above reasons, further research is needed into the differences between women and men and into how romantic relationships shape their perceptions of stress and pandemic-related difficulties. Future research should also focus on comparing coupled and unpartnered individuals and look into other than heterosexual relationships.

Given that the respondents were patients—i.e., individuals receiving treatment—it is worth noting that the COVID-19 pandemic and the accompanying lockdowns limited out-of-home leisure opportunities, forcing people to stay confined (together) at home most of the time, and that this may have decreased overall satisfaction with life [107,108], which in turn may have affected peoples’ relationships with their romantic partners. Moreover, additional challenges and difficulties emerged in relationships, where one partner had a chronic disease, a circumstance leading to reduced mental well-being and increased anxiety and worries about their family members being infected with COVID-19. A family member having a disease was associated with increased anxiety related to COVID-19, depression and stress, which affected relationship quality in couples [109].

Given the knowledge available, this research is the first to analyze both perceived stress and pandemic-related difficulties during the COVID-19 pandemic in Poland. However, this study has some limitations. First, due to the limitations regarding the questionnaire length, the study has few items assessing mental well-being, types of stress or sexual life; therefore, future research should include a more in-depth analysis of the above aspects. This study measures stress with respect to four statements; however, stress is multidimensional (e.g., daily, chronic) and significant dimensions may vary across individuals. We lack information on how respondents cope with stress or on their stress experiences early on in their lives. Moreover, the study collected data at the final stage of the COVID-19 pandemic and it did not include longitudinal data. Stress levels might have been higher following the pandemic outbreak compared to the time when the respondents had already adapted—even if only partially—to the new circumstances. Moreover, the degree to which couples are resilient, maintain positive functioning of their relationship and stay together may vary depending on additional factors. Furthermore, the nature of an online survey allows the generalization of the study findings to a population with higher digital competence.

## 5. Conclusions

The study conducted an aggregated analysis of romantic relationship variables and determined the most significant association paths among the variables under study in a comprehensive stress model. The findings prove that women were characterized by higher levels of stress and pandemic-related difficulties and less favorable assessments of their relationships. Furthermore, the assessment of the relationship quality, attachment styles and other sociodemographic variables affected the variability of perceived stress and other pandemic-related difficulties. For women, having children and their financial situation were significant predictors of perceived stress and this allowed us to understand the differences between men and women, which, in turn, will help to adopt clinical recommendations regarding coping with major stressors when faced with a pandemic. Moreover, the relationship attachment styles were directly and indirectly related to stress through an emotional reaction to the pandemic. The tested model highlights the key predictors of stress, not only allowing us to better understand the impact of the COVID-19 pandemic on the functioning of women and men, but also presenting the conclusions relevant for clinicians and therapists working with both individuals and couples in times of severe stress. It is necessary to take action to promote the survival of global crises and strengthen mental health resources so that strategies for coping with further difficulties can be developed.

## Figures and Tables

**Figure 1 ijerph-20-02988-f001:**
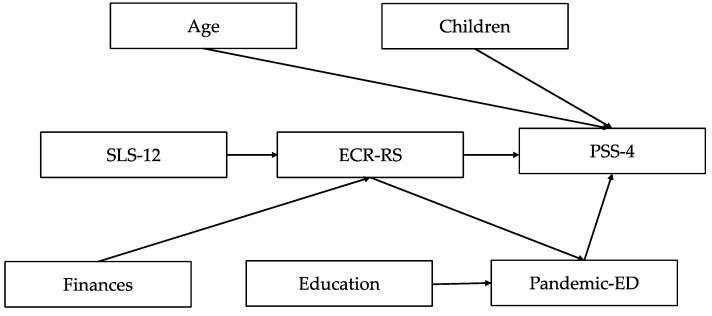
The theoretical framework of the SEM model.

**Figure 2 ijerph-20-02988-f002:**
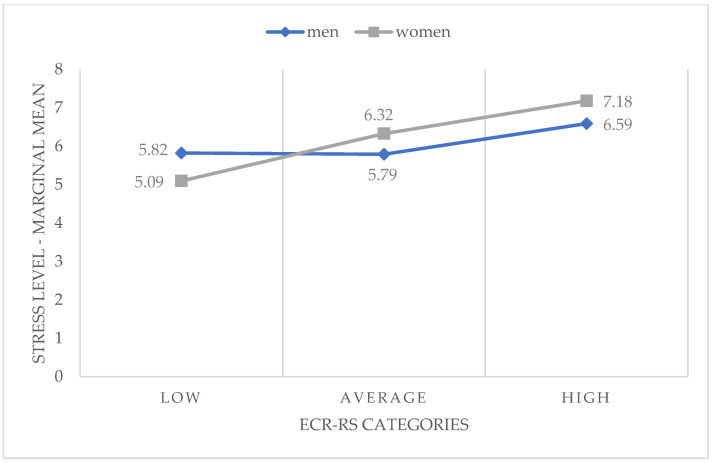
Interaction between gender and the assessment of relationship attachment styles (ECR-RS categories) as predictors of perceived stress level in women and men (N = 1405) (marginal means from the GLM model).

**Figure 3 ijerph-20-02988-f003:**
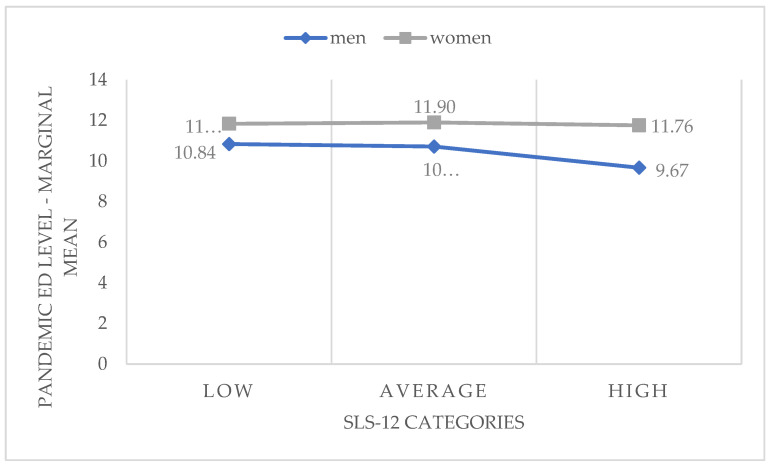
Interaction between gender and the assessment of relationship quality (SLS-12 categories) as predictors of perceived pandemic-related difficulties in women and men (N = 1405) (marginal means from GLM model).

**Table 1 ijerph-20-02988-t001:** Sample characteristics: individuals in relationships (all data presented as percentages).

Variable	Categories	Total N = 1405	Male N = 709	Female N = 696
Education	Elementary and vocational	25.2	20.6	29.9
	High school	35.2	33.9	36.6
	University	39.6	45.6	34.5
Place of residence	Large cities	30.6	36.8	24.3
	Small towns	31.2	33.4	29.0
	Rural areas	38.2	29.9	46.7
Relationship status	Formal	77.5	78.7	76.3
	Informal	22.5	21.3	23.7
Children	Yes	83.6	83.2	84.1
	No	16.4	16.8	15.9
Financial situation	Worse	36.2	33.9	38.6
	Same or hard to say	58.9	60.9	56.9
	Improved	4.8	5.2	4.5

**Table 2 ijerph-20-02988-t002:** Fitness Model Summary (N = 1405).

CMIN/DF	RMR	GFI	AGFI	CFI	NFI	RFI	IFI	TLI	RMSEA
2.432/1	0.008	0.999	0.991	0.999	0.999	0.994	0.999	0.996	0.032

CMID/DF: the Chi-square value/ degrees of freedom; RMR: root mean square residual; GFI: goodness-of-fit index; AGFI: adjusted goodness-of-fit index; CFI: comparative fit indices; NFI: normed fit index; IFI: incremental fit index; TLI: Tucker-Lewis’s index; RMSEA: root mean square error of approximation.

**Table 3 ijerph-20-02988-t003:** Descriptive statistics, floor and ceiling effects for individual items (N = 1405).

Pandemic-ED Items	M	SD	Skewness	Kurtosis	Floor Effect (%)	Ceiling Effect (%)
frustration	3.03	1.164	−0.056	−0.599	12.6	12.7
loneliness	2.38	1.135	0.471	−0.454	27.8	5.4
anger	2.81	1.204	0.113	−0.760	17.7	10.5
anxiety	2.76	1.139	0.175	−0.512	16.2	9.0

M: mean; SD: standard deviation.

**Table 4 ijerph-20-02988-t004:** Correlational analyses PSS-4, Pandemic-ED, ECR-RS total and SLS-12 (rho Spearman) (N = 1405).

Scale	PSS-4	Pandemic-ED	ECR-RS Total	SLS-12	Age in Years
PSS-4	1	0.444 **	0.370 **	−0.248 **	−0.232 **
Pandemic-ED	0.444 **	1	0.262 **	−0.226 **	−0.084 *
ECR-RS-total	0.370 **	0.262 **	1	−0.701 **	−0.098 **
SLS-12	−0.248 **	−0.226 **	−0.701 **	1	−0.051
Age in years	−0.232 **	−0.084 *	−0.098 **	−0.051	1

** *p* < 0.001; * *p* < 0.01; PSS-4: Perceived Stress Scale; Pandemic-ED: Pandemic Emotional Difficulties Scale; ECR-RS: Experiences in Close Relationships-Revised Scale; SLS-12: Short Love Scale.

**Table 5 ijerph-20-02988-t005:** Differences between women and men within the scope of the study scales (N = 1405).

Scale	M (SD) (N = 1405)	Men (N = 709)	Women (N = 696)	U Mann-Whitney	*p*	Glass’ Delta
PSS-4	5.65 (3.17)	5.15	6.16	−5.741	<0.001	0.331
Pandemic-ED	10.99 (3.83)	10.12	11.86	−8.720	<0.001	0.474
ECR-RS_total	22.52 (11.26)	21.70	23.35	−1.862	0.068	
Anxiety	7.46 (4.84)	7.05	7.87	−2.371	0.018	0.183
Avoidance	15.06 (8.17)	16.65	15.48	−0.607	0.544	
SLS-12	48.46 (10.23)	49.35	47.54	−2.564	0.010	0.192
Sexual life	7.21 (2.38)	7.27	7.16	−0.618	0.537	
Closeness	7.77 (2.09)	7.90	7.64	−1.434	0.152	
Commitment	33.47 (6.62)	34.18	32.74	−3.386	<0.001	0.215

M: mean; SD: standard deviation; U: Mann–Whitney test value; *p*: the significance value; Glass’ delta: effect size for statistically important differences; PSS-4: Perceived Stress Scale; Pandemic-ED: Pandemic Emotional Difficulties Scale; ECR-RS: Experiences in Close Relationships-Revised Scale; SLS-12: Short Love Scale.

**Table 6 ijerph-20-02988-t006:** Hierarchical regression for the PSS-4 variable. Variables in the model (N = 1405).

Independent Variable PSS-4	Model 1		Model 2		Model 3	
	β	*p*	β	*p*	β	*p*
Gender *	0.094	<0.001	0.099	<0.001	0.027	0.262
Age	−0.165	<0.001	−0.138	<0.001	−0.143	<0.001
Education *	−0.026	0.307	−0.021	0.402	−0.043	0.060
Place of residence *	0.009	0.717	−0.005	0.829	0.000	0.991
Relationship status *	−0.022	0.465	−0.007	0.823	0.000	0.988
Length of relationship	−0.041	0.353	−0.032	0.456	−0.017	0.668
Children *	−0.034	0.223	−0.035	0.208	−0.047	0.068
Financial situation *	0.159	<0.001	0.145	<0.001	0.053	0.024
SLS-12	−0.250	<0.001	−0.045	0.212	−0.025	0.453
ECR-RS			0.280	<0.001	0.219	<0.001
Pandemic-ED					0.358	<0.001
R-sq	0.163		0.198		0.298	

* dummy variable. β: beta coefficient; *p*: the significance value; PSS-4: Perceived Stress Scale; SLS-12: Short Love Scale; ECR-RS: Experiences in Close Relationships-Revised Scale; Pandemic-ED: Pandemic Emotional Difficulties Scale; R-sq: coefficient of determination.

**Table 7 ijerph-20-02988-t007:** Hierarchical regression for pandemic-related emotional difficulties. Variables in the model (N = 1405).

Independent Variable Pandemic-ED	Model 1		Model 2		Model 3	
	β	*p*	β	*p*	β	*p*
Gender *	0.199	<0.001	0.202	<0.001	0.166	<0.001
Age	−0.005	0.910	−0.011	0.789	0.062	0.118
Education *	0.060	0.016	0.063	0.011	0.071	0.002
Place of residence *	0.016	0.513	0.014	0.572	0.012	0.601
Relationship status *	−0.029	0.334	−0.020	0.511	−0.017	0.536
Length of relationship	−0.047	0.285	−0.042	0.340	−0.030	0.463
Children *	0.034	0.221	0.034	0.221	0.047	0.072
Financial situation *	0.266	<0.001	0.256	<0.001	0.203	<0.001
SLS-12	−0.180	<0.001	−0.055	0.131	−0.038	0.257
ECR-RS			0.171	<0.001	0.068	0.048
PSS_4					0.367	<0.001
R-sq	0.159		0.172		0.280	

* dummy variable. β: beta coefficient; *p*: the significance value; Pandemic-ED: Pandemic Emotional Difficulties Scale; SLS-12: Short Love Scale; ECR-RS: Experiences in Close Relationships-Revised Scale; PSS-4: Perceived Stress Scale; R-sq: coefficient of determination.

**Table 8 ijerph-20-02988-t008:** Standardized regression weights for SEM models.

	Total (N = 1405)		Men (N = 709)		Women (N = 696)	
Path	Estimate	*p*	Estimate	*p*	Estimate	*p*
SLS-12 -> ECR-RS	−0.724	<0.001	−0.701	<0.001	−0.739	<0.001
Finances -> ECR-RS	0.056	0.002	0.051	0.053	0.061	0.015
ECR-RS -> Pandemic-ED	0.228	<0.001	0.279	<0.001	0.170	<0.001
Education -> Pandemic-ED	0.038	0.123	0.057	0.103	0.067	0.061
ECR-RS -> PSS-4	0.241	<0.001	0.219	<0.001	0.266	<0.001
Pandemic-ED -> PSS-4	0.378	<0.001	0.371	<0.001	0.360	<0.001
Children -> PSS-4	0.049	0.042	0.016	0.650	0.094	0.005
Age -> PSS-4	−0.163	<0.001	−0.114	0.002	−0.196	<0.001

*p*: the significance value; Pandemic-ED: Pandemic Emotional Difficulties Scale; SLS-12: Short Love Scale; ECR-RS: Experiences in Close Relationships-Revised Scale; PSS-4: Perceived Stress Scale.

**Table 9 ijerph-20-02988-t009:** Standardized indirect effects with 95% confidence intervals (CIs) (bootstrap sample = 5000).

	Total (N = 1405)		Men (N = 709)		Women (N = 696)	
Path	Indirect Effect (95%CI)	*p*	Indirect Effect (95%CI)	*p*	Indirect Effect (95%CI)	*p*
Finances -> Pandemic-ED	0.013 (0.006–0.021)	0.002	0.014 (0.002–0.028)	0.064	0.010 (0.003–0.020)	0.010
Finances -> PSS-4	0.018 (0.008–0.029)	0.002	0.016 (0.002–0.032)	0.068	0.020 (0.006–0.035)	0.013
SLS-12 -> Pandemic-ED	−0.165 (−0.198–−0.131)	<0.001	−0.196 (−0.240–−0.151)	<0.001	−0.126 (−0.172–−0.078)	<0.001
SLS-12 -> PSS-4	−0.236 (−0.270–−0.204)	<0.001	−0.226 (−0.268–−0.183)	<0.001	−0.242 (−0.292–−0.193)	<0.001
Education -> PSS-4	0.015 (0.000–0.030)	0.111	0.021 (0.000–0.042)	0.096	0.024 (0.003–0.046)	0.055
ECR-RS -> PSS-4	0.086 (0.068–0.105)	<0.001	0.103 (0.078–0.135)	<0.001	0.061 (0.039–0.085)	<0.001

*p*: the significance value; Pandemic-ED: Pandemic Emotional Difficulties Scale; SLS-12: Short Love Scale; ECR-RS: Experiences in Close Relationships-Revised Scale; PSS-4: Perceived Stress Scale.

## Data Availability

The data are owned by Warsaw University and are not to be made freely publicly available.

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
