# Peer review of "Pandemic-Related Stress and Other Emotional Difficulties in a Sample of Men and Women Living in Romantic Relationships during the COVID-19 Pandemic"

_ijerph, 2023, doi:10.3390/ijerph20042988_

Round 1

Reviewer 1 Report

Review of “Pandemic-related stress and other emotional difficulties in a 2 sample of men and women living in romantic relationships 3 during the COVID-19 pandemic” submitted to International Journal of Environmental Research and Public Health

The manuscript consists of an empirical analysis of how the COVID-19 pandemic affected the lives of people engaged in romantic relationships. Among the strong points of the manuscript is that the research is based on a large sample of respondents and the authors conduct several multivariate statistical analyses, including regression modelling and structural equation modelling (SEM). However, there are also some important issues that should be carefully addressed by the authors, before reconsidering this paper for publication.

1) The manuscript lacks a proper theoretical discussion of one of the key terms featuring in its very title, that is, “romantic relationships.” Although the paper is written from the disciplinary perspective of psychology, I suggest the authors to consider developing their theoretical section towards a perspective grounded in the sociology of the family in general, and the sociology of love and romantic relationships in particular.

2) Methodologically, I would point out two significant issues, one regarding the hypotheses, the other concerning the sample.

            2.1) Hypotheses: the authors formulated four hypotheses, which are specified at the end of the Introduction. I believe that none of these four hypotheses are stated in a proper form. By proper form, I mean satisfying two conditions: a) to mention a relationship between two variables (which they do); and b) to specify the direction of the relationship, more precisely, how exactly does X influence Y (which they do not). For instance, H1 states that “Women and men differ in how they perceive stress, experience emotional difficulties, and how they assess the relationship and attachment style.” It is not enough to state that women and men differ in one regard or another. The authors should also indicate in what sense or direction women and men differ. This observation holds true for all four hypotheses. Additionally, H3 and H4 are poorly formulated, in that they are rather unclear. For instance, what do the authors mean by “directly and indirectly” in H3?

            2.2) Sample: the manuscript mentions that the sample size is N = 2050 and that “the sample accurately represents the population.” However, the authors should specify if the sample is probabilistic or non-probabilistic. If it is a probabilistic sample, they should further specify the error margin of the sample. If it is a non-probabilistic sample, the authors should be transparent about this and assume it as a methodological limit.

3) In the Results section, it is not clear whereas the four hypotheses were validated or rejected by the statistical findings reported here. This section should be thoroughly restructured around testing the empirical adequacy of each of the four hypotheses, so that the results to pinpoint exactly if they are supported or not by the findings.

4) The Conclusions make no mention of the limitations of the study. I suggest the authors to use this section for acknowledging any theoretical, methodological, and/or analytical limitations of their approach.

5) The entire manuscript requires a thorough professional proofreading as well as editing. Some sentences do not sound right, and the text is filled with typos are other grammatical errors.

Author Response

Thank you very much for all your comments. The responses are attached. 

Reviewer 2 Report

The article is very interesting and well done. The methodological part is accurate and correct. The results are also interesting, although it is not made sufficiently clear what the point of knowing all this is. 

It is therefore recommended:

- In the introduction it would be useful to mention a European study carried out to define the coping strategies used by the population, to include at least an indication of how people tried to cope with the pandemic. Since research often uses the construct of coping, it is certainly important to consider this study;

- Separate the part on objectives and hypotheses from the introduction and describe it in more detail. Before the hypothesis, better specify the objectives (which are different from the hypothesis, i.e. to make it clear why the study was carried out and what it is for). In this section, before the objectives, clearly specify the research question.

- Revise the English (some typos appear: e.g. "anal-ysis, line 216);

- In the discussion part, take the research question and objectives and relate them to the results.

Author Response

(The authors gave the same response as above.)

Round 2

Reviewer 1 Report

I am pleased to see that the authors have taken care to address in a satisfactory manner all the issues that were pointed out in the peer-review report. I believe that the paper is now is a better shape and that it can be published as it stands now.